# Using *Acanthamoeba* spp. as a cell model to evaluate *Leishmania* infections

**Helena Lúcia Carneiro Santos** [1,2,3]*, **Gabriela Linhares Pereira**[1], **Rhagner Bonono do Reis**[1], **Igor Cardoso Rodrigues**[1], **Claudia Masini d'Avila**[1], **Vitor Ennes Vidal**[1]

**1** Laboratório de Doenças Parasitárias, Instituto Oswaldo Cruz/FIOCRUZ, Rio de Janeiro, Rio de Janeiro, Brazil, **2** Coleção de Protozoários da FIOCRUZ, Instituto Oswaldo Cruz/FIOCRUZ, Rio de Janeiro, Rio de Janeiro, Brazil, **3** Laboratório de Parasitologia, Departamento de Patologia da Faculdade de Medicina da Universidade Federal Fluminense, Rio de Janeiro, Rio de Janeiro, Brazil

* helenalucias@ioc.fiocruz.br

## Abstract

Leishmaniasis represents a severe global health problem. In the last decades, there have been significant challenges in controlling this disease due to the unavailability of licensed vaccines, the high toxicity of the available drugs, and an unrestrained surge of drug-resistant parasites, and human immunodeficiency virus (HIV)–*Leishmania* co-infections. *Leishmania* spp. preferentially invade macrophage lineage cells of vertebrates for replication after subverting cellular functions of humans and other mammals. These early events in host–parasite interactions are likely to influence the future course of the disease. Thus, there is a continuing need to discover a simple cellular model that reproduces the *in vivo* pathogenesis. *Acanthamoeba* spp. are non-mammalian phagocytic amoeba with remarkable similarity to the cellular and functional aspects of macrophages. We aimed to assess whether the similarity reported between macrophages and *Acanthamoeba* spp. is sufficient to reproduce the infectivity of *Leishmania* spp. Herein, we analyzed co-cultures of *Acanthamoeba castellanii* or *Acanthamoeba polyphaga* with *Leishmania infantum*, *Leishmania amazonensis*, *Leishmania major*, and *Leishmania braziliensis*. Light and fluorescence microscopy revealed that the flagellated promastigotes attach to the *A. castellanii* and/or *A. polyphaga* in a bipolar and or random manner, which initiates their uptake via pseudopods. Once inside the cells, the promastigotes undergo significant changes, which result in the obligatory amastigote-like intracellular form. There was a productive infection with a continuous increase in intracellular parasites. However, we frequently observed intracellular amastigotes in vacuoles, phagolysosomes, and the cytosol of *Acanthamoeba* spp. Our findings corroborate that *Leishmania* spp. infects *Acanthamoeba* spp. and replicates in them but does not cause their rapid degeneration or lysis. Overall, the evidence presented here confirms that *Acanthamoeba* spp. have all prerequisites and can help elucidate how *Leishmania* spp. infect mammalian cells. Future work exposing the mechanisms of these interactions should yield novel insights into how these pathogens exploit amoebae.

**Data Availability Statement:** All relevant data are within the paper and its Supporting Information files.

**Funding:** This study was supported by the Instituto Oswaldo Cruz intramural funding (PAEF II-IOC-23-FIO-18-2-53), and the Fundação Carlos Chagas Filho de Amparo à Pesquisa do Estado do Rio de Janeiro (FAPERJ) Project ID: N° DO PROCESSO SEI-260003/002148/2021 - APQ1. The funders had no role in study design, data collection and analysis, decision to publish, or preparation of the manuscript.

**Competing interests:** The authors have declared that no competing interests exist.

## Author summary

Leishmaniasis represents a severe global health problem, and drug resistance is a growing concern. *Leishmania* spp. are obligate intracellular parasites that survive within cells of the vertebrate macrophage, modulating their activation. Understanding the multilayered relationship between metabolism and function of innate immune cells during infection has great therapeutic and preventive potential. Mammalian macrophages and *Acanthamoeba* spp. display similarities in the molecular mechanisms involved in directional motility, recognition, binding, engulfment, and the phagolysosomal processes, and they express similar receptors. Hence, we hypothesize that *Acanthamoeba* spp. represent a model that can be used to evaluate macrophage–pathogen interactions from the perspective of innate immunity. However, it has not yet been described whether *Leishmania* spp. can survive in *Acanthamoeba* spp. We found that *Acanthamoeba* spp. support *Leishmania* growth. *Acanthamoeba* spp. contain ancient pathogen recognition mechanisms, and *Leishmania* spp. could manipulate amoeboid functions to favor their survival and replication based on strategies of ancestral metabolic pathways. Our robust evidence highlights that amoebae could be used as a model to understand the biology and evolution of host–*Leishmania* interactions.

## Introduction

Leishmaniasis is a term used to define a wide-ranging group of diseases caused by protozoa of the genus *Leishmania* [1,2]. These infections result in a broad spectrum of clinical manifestations that range from asymptomatic infection to localized, disseminated ulcers, sometimes with mucosal involvement (cutaneous leishmaniasis), or affect internal organs, such as the liver, spleen, and bone marrow (visceral leishmaniasis), which can be fatal if left untreated [3,4]. In recent years, leishmaniasis has been considered a neglected and emerging group of diseases. These diseases constitute a global grave public-health problem that leads to great suffering and high socio-cultural and economic costs [5]. There is no available vaccine and drug resistance is emerging [6]. Additionally, the available drugs have severe side effects, further complicating treatment, and cannot eliminate the pathogen [6]: The parasite remains in the body and can cause a relapse when there is immunosuppression [6,7]. According to Word Health Organization (WHO), over 1 billion people are at risk of infection in high-burden countries [8,9]. There are an estimated 700,000 to 1 million new cases each year, highlighting the importance of this public health problem [10].

 In a vertebrate host, *Leishmania* parasites can evade the defense barriers of the host and even benefit from them [9]. Indubitably, macrophages are key cells to the body's guards against pathogens [10]. However, *Leishmania* spp. have evolved to modulate vacuole biogenesis to generate a unique intravacuolar niche that is suitable for their proliferation inside macrophages [4,11]. In this niche, *Leishmania* differentiates into the amastigote form that inhabits digestive vacuoles, which fuse to macrophage lysosomes, generating phagolysosomes [12]. Elucidating the complex *Leishmania*–macrophage interactions is crucial to prevent and treat leishmaniasis [11,12]. To survive successfully and multiply within macrophages, *Leishmania* must undergo profound biochemical and morphological adaptations to enter a host cell, establish an infection, and evade or modulate the subsequent immune response [13,14]. These abilities might have resulted from cellular and molecular adaptations to the ancestral phagocytosis process [15].

Mammalian macrophages and *Acanthamoeba* spp. display similarities in the molecular mechanisms involved in directional motility, recognition, binding, engulfment, and phagolysosome processing of bacteria and fungi [16]. They also display conserved pathogen-recognition mechanisms [16,17]. *Acanthamoeba* spp. are opportunistic protozoans, ubiquitously distributed in nature, that are mainly found in soil and water. In general, the life cycle of these amoeba consists of two stages, trophozoite and cyst. Trophozoites are the motile forms that mainly feeds on bacteria, algae, yeast, or small organic particles via phagocytosis or pinocytosis [16,18]. Trophozoites closely resemble the internal structure of a mammalian cell. They contain various organelles such as mitochondria, ribosomes, centrosomes, vacuoles, and the Golgi apparatus. Vacuoles are a critical component of trophozoites: they include the contractile vacuole, which expels water for osmotic regulation, lysosomes, digestive vacuoles, and many glycogen-containing vacuoles [18].Over the past several decades, *Acanthamoeba* spp. have gained increasing attention due to their ability to act as a host/reservoir for microbial pathogens, particularly their role in causing damage and sometimes fatal human and animal infections [16]. Recent studies suggest that *Acanthamoeba*–microbe adaptations are considered an evolutionary model of macrophage–pathogen interactions and acts as a "genetic melting pot" with cross-species exchange of genes [17].

Due to their remarkable cellular and functional similarities, authors have speculated that *Acanthamoeba* spp. and macrophages are evolutionarily related [15,18]. An exciting hypothesis is *Acanthamoeba* spp. may be an *in vitro* model to investigate pathogen–macrophage interactions [19].

Phagocytosis is a complex process that has been maintained during evolution [15,20]. *Acanthamoeba* spp. are well known for their exuberant motility and active pseudopodia when interact with bacteria, viruses, protists, and fungi [17,19]. *Acanthamoeba* spp. can harbor important human pathogens such as *Legionella pneumophila*, *Aeromonas* spp., *Pseudomonas* spp., *Escherichia coli*, and *Mycobacterium* spp., and others cultured from clinical specimens [19,21–23]. It is noteworthy that some of these pathogens can subvert the microbicidal activity and replicate inside *Acanthamoeba* spp. with mechanisms that are similar to those used for survival within macrophages [24–26]. While *Acanthamoeba* spp. and macrophages take up microbes via phagocytosis, whether these distinct hosts have an evolutionary or convergent evolution remains unknown. Moreover, not all the phagocytic pathways are totally conserved between amoebae and macrophages, the endocytic system evasion mechanisms such as inhibition of phagosome–lysosome fusion, are common to escape from either host [19].

Despite efforts to unravel the mechanisms of *Leishmania* pathogenicity and the risk of infection, and to develop novel treatments and vaccines against the parasite, there are still gaps in the state-of-the-art that need be explored. The innate immunity pathways are widely conserved in nature, and evaluation of *Acanthamoeba*–*Leishmania* interactions could represent an important model to understand the mechanisms of virulence and host immunity. We evaluated whether *Acanthamoeba* spp. are similar enough to macrophages to reproduce *Leishmania* spp. infectivity. Understanding these interactions will potentially reveal new mechanisms for investigating host–parasite interactions.

## Material and Methods

### Schematic overview of *Leishmania-Acanthamoeba* interaction assay

**Acanthamoeba stains and cultivation conditions.** Two *Acanthamoeba* spp. strains were used in this study: an environmental strain *Acanthamoeba castellanii* designated as Neff (ATCC 30010), and *Acanthamoeba polyphaga* (ATCC 30431) that was isolated from corneal scrapings of a patient with acute ulceration of the right eye. The amoebae were grown in

Peptone Yeast Glucose (PYG) medium containing 0.75% (w/v) proteose peptone, 0.75% (w/v) yeast extract, and 1.5% (w/v) glucose as a monolayer in 25 cm$^2$ tissue culture flasks at 27˚C. When cells formed a monolayer, the trophozoites were harvested, centrifuged (200 *g* for 5 min), washed three times with phosphate-buffered saline (PBS), and suspended in PYG broth or PBS. Cells were counted in a hemocytometer chamber, and the viability was assessed by the trypan blue exclusion assay. Prior to the experiments, the amoebae were grown in tissue culture flasks at 27˚C without shaking, and the media refreshed after 17–20 h. In addition, the influence of culture media (Grace's insect cell culture medium (Grace's), liver infusion tryptose [LIT], and Schneider's insect medium [Schneider's], without supplementation with fetal calf serum [FCS]) on the growth and viability of *Acanthamoeba* spp. was assessed.

*Acanthamoeba* spp growth was monitored daily for 6 days with a Neubauer chamber. Viability was assessed by mobility and lack of staining after challenging with trypan blue. All results correspond to three independent assays that were run in duplicates. Before starting the experiments, parasites were maintained in each medium for a minimum of three passages.

**Leishmania strains and cultivation conditions of promastigotes.** Three causative species of cutaneous leishmaniasis, *Leishmania amazonensis* (MHOM/BR/PH8), *Leishmania braziliensis* (strain Thor, MCAN/BR/1998/619), and *Leishmania major* (MHOM/SU/1973/5-ASKH), and one causative species of visceral leishmaniasis, *Leishmania infantum* (MHOM/BR/1974/PP75), were used in this study. Promastigotes were obtained from the Protozoa Culture Collection (COLPROT) and the *Leishmania* Collection (CLIOC) from Instituto Oswaldo Cruz–Fundação Oswaldo Cruz (IOC-FIOCRUZ) and cultured in Schneider's medium (pH 7.2) supplemented with 20% fetal bovine serum (FBS), streptomycin (40 μg/mL), and penicillin (100 U/mL) at 27˚C. The parasites were maintained by serial passaging twice per week. Before starting the experiments, promastigotes were sub-passaged to a fresh medium every 4 days to maintain the growth and viability of the parasites.

**Co-culture of Leishmania spp. and Acanthamoeba spp.** For light microscopy analyses, *A. castellanii* and *A. polyphaga* were seeded in eight-well chamber Lab-Tek chambers (Nunc, Roskilde, Denmark) at $1 \times 10^5$ cells per well. Then, $1 \times 10^6$ promastigotes (*L. braziliensis*, *L. amazonensis*, *L. infantum*, and *L. major*) in the stationary growth phase of the respective culture media (1:10 ratio) were added to two wells each at 37˚C or 27˚C. After 3, 6, 12, 24, 36, 72 h, and 96 h, the supernatant from each chamber slide well was discarded and the top of the plate was removed. Non-adhered promastigotes were removed by three washing cycles with PBS (pH 7.2). The attached cells were fixed with methanol before being submitted to panoptic fast staining. To evaluate infectivity, the slides were analyzed under an optical microscope (Olympus Optical) at 1000× total magnification using immersion oil. For fluorescence microscopy analyses, promastigotes in the mid-log growth phase were incubated with 0.1% fluorescein isothiocyanate (FITC) in a carbonate-bicarbonate buffer (pH 9.4; 9.5 mL of 0.2 M Na$_2$CO$_3$ mixed with 41.5 mL of 0.2 M NaHCO$_3$) for 1 h at room temperature in the dark, based on a previously described method [27]. Finally, the pellet was resuspended in 1 mL of PBS. *A. castellanii* and *A. polyphaga* were seeded in eight-well chamber Lab-Tek chambers (Nunc, Roskilde, Denmark) at a density of $1 \times 10^5$ cells per well. Then, $1 \times 10^6$ FITC-labeled promastigotes forms (*L. braziliensis*, *L. amazonensis*, *L. infantum*, and *L. major*) in the stationary growth phase of the respective culture media (1:10 ratio) were added to two wells each, as described above. *Each* dried *slide was* fixed with absolute methanol and examined under a fluorescence microscope (Zeiss Axiolab) at 400× and 1000× total magnification using immersion oil (Fig 1).

**The parasite release and differentiation assay.** Twenty-four hours after infection, *Acanthamoeba* monolayers were washed and incubated with 0.01% sodium dodecyl sulfate (SDS; Sigma-Aldrich) for 10 min to allow lysis of *Acanthamoeba* and the release of

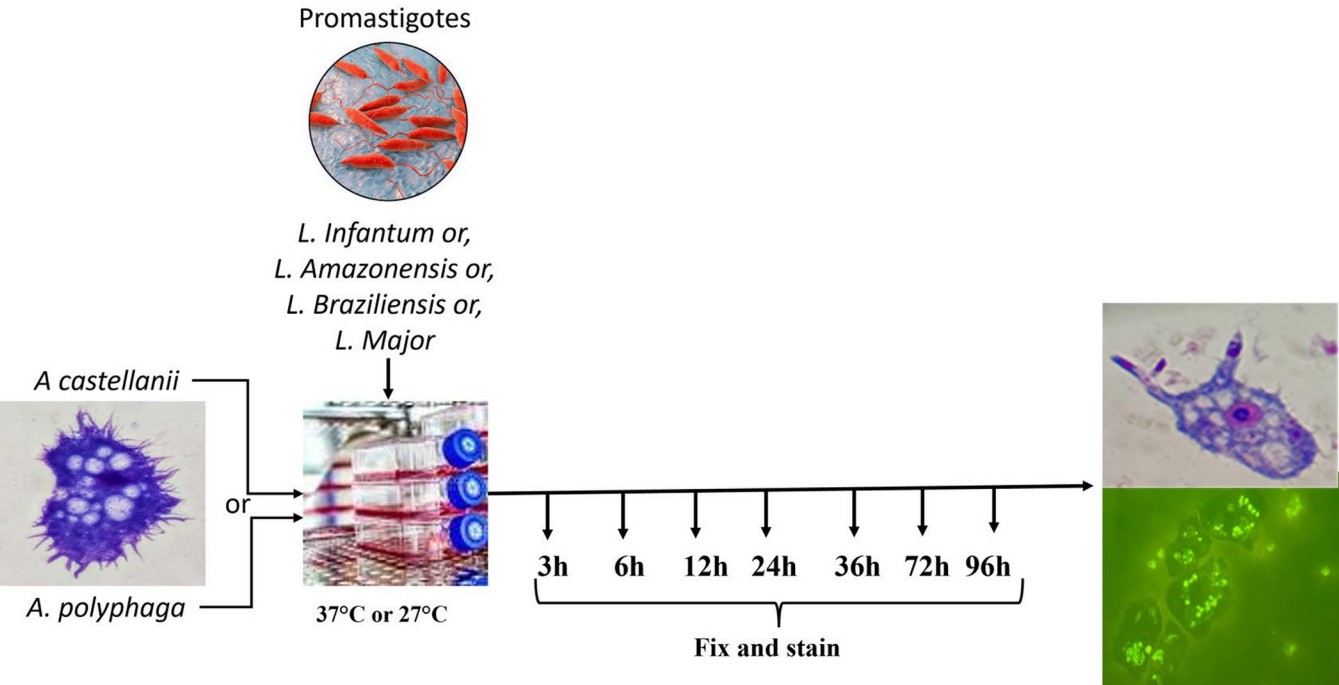

**Fig 1. Schematic overview of *Leishmania-Acanthamoeba* interaction assay.** The host cells (*Acanthamoeba* castellanii or *Acanthamoeba* polyphaga) were plated, then infected by Leishmania spp promastigotes transformed in amastigotes of each species with a multiplicity of infection (MOI) of 1:10. Three, six, 12, 24, 36, 72 h, and 96 h post-infection (HPI), cells were fixed and stained with Panoptic fast staining and visually analyzed. These observations were further confirmed by fluorescence microscopic imaging of macrophages infected with FITC-labeled *Leishmania* spp.

amastigotes. Then, cultures were fed with five mL of Schneider's medium, supplemented with 10% FBS, and incubated at 27˚C and for an additional 2 days.

## Results

### The impact of traditional media on the growth of *Acanthamoeba castellanii*

Before the infection experiments, *Acanthamoeba* spp. were assessed for growth potential in conventional culture media commonly used to isolate and maintain *Leishmania* spp. *in vitro*, namely Liver Infusion Typtose (LIT), Schneider's medium, and Grace's medium.

*Acanthamoeba* spp. were able to tolerate different growth conditions, without special nutritional requirements. Over six days, we counted the number of amoebae that grew in each culture medium (Fig 2).

At the control medium, trophozoite reached the beginning of the logarithmic phase (BLP) on day two, and a decrease in cell density was observed at six days throughout the experiment.

An initial population of 1.0 x 106 organisms/ml regularly gave rise to a maximal population.

1.0 X 107 organisms/ml within five days. *A. castellanii* showed slow growth over six days in LIT and Schneider's media (1.3 ×106/mL). The number of parasites was low compared with the classic PYG medium (1.8 × 107/mL at five days); however, morphological homogeneity and the trophozoites remained viable with a slightly increased cellular count at six days. It was quickly observed live trophozoites were extremely pleomorphic, moderately mobile, cytoplasmic acantopodia of varying size and shape, and large round to oval nuclei with prominent and centrally located nucleolus as well as abundant vacuoles of different sizes and content.

**Fig 2. Effect of media on the growth of *Acanthamoeba castellanii*.** Number of trophozoites in the different growth media: PYG, black, as the control media; LIT (yellow); Schneider's (green), and Grace's medium (red). Black arrowheads = the beginning of the logarithmic phase (BLP). **The days of quantification are represented on the x-axis and the concentration of parasites per mL is expressed in a logarithmic scale on the y-axis.**

We assessed parasite viability with the trypan blue exclusion assay and by observing pseudopodia and morphology. The trypan blue exclusion assay showed a smooth reduction of viable parasites compared with the controls. Except for Grace's medium, the other media tested showed a similar viability percentage. The estimated viability of *Acanthamoeba* was not less than 99%.

## *Acanthamoeba–Leishmania* co-culture

We co-cultured *L. braziliensis*, *L. amazonensis*, *L. infantum*, and *L. major* with *A. castellanii* or *A. polyphaga* for up to 96 h. Light microscopical examination showed, after initial contact with the flagellum, promastigote forms attached to *Acanthamoeba* cells.

Formation of surface projections of the amoeba and protrusions engulfed the promastigotes from the flagellar tip toward the cell body (Fig 3A, 3B, and 3D). We noted that long tubular pseudopods tightly encircled their flagellum (Fig 3C and 3E) or the posterior (non-flagellated) pole (Fig 3C). Note the pseudopodial membrane that appears to form a tight grip on the promastigotes and to "flow" forward to engulf them. Engulfment of the adhering promastigotes occurred via funnel-like extensions of the *Acanthamoeba* surface extending along the radiating parasites (Fig 3C and 3F). Insertion of promastigote flagellum into the pseudopodium of an *A. castellanii*, indicative of a mode of entry with the flagellar end first.

## *Acanthamoeba–Leishmania* co-culture at three hours of interaction

Once inside the cells, the promastigotes undergo significant changes, which result in the obligatory amastigote-like intracellular form. *Acanthamoeba* spp. engulfed promastigotes,

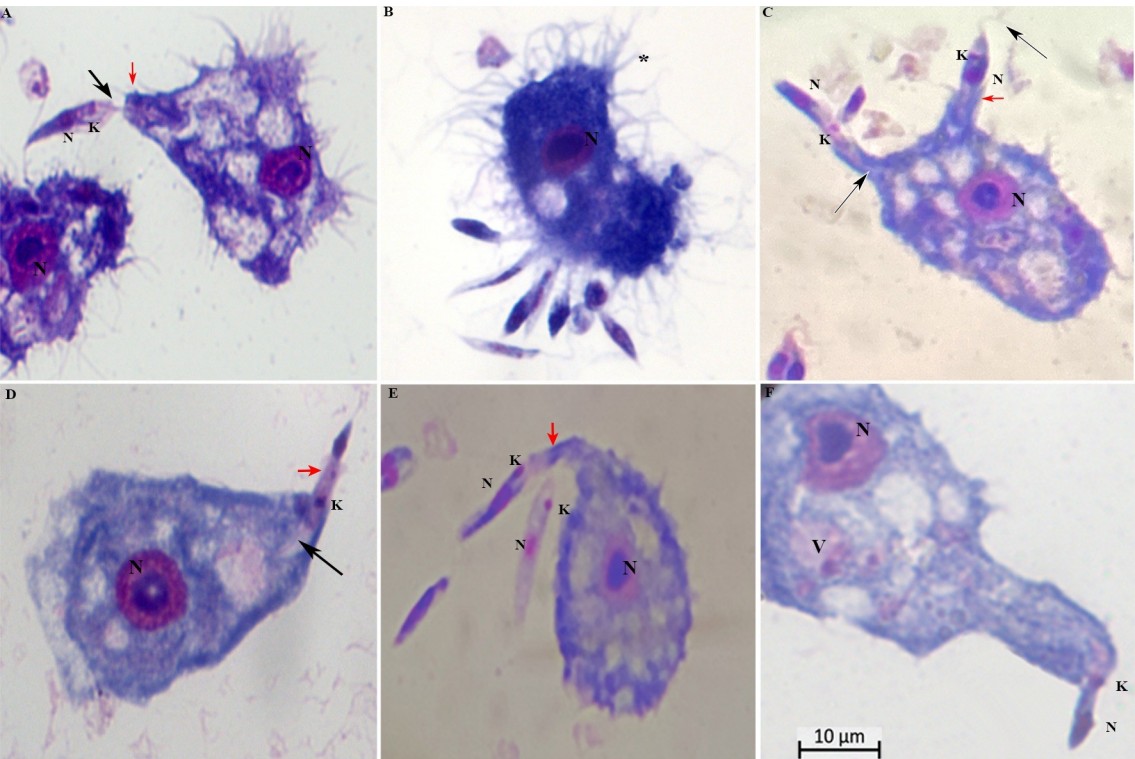

**Fig 3. Photomicrographs of *Acanthamoeba castellanii* infected with *Leishmania* spp. stained with Panoptic fast staining.** Parasite ratio of 1:10. Protrusions engulfed the promastigotes from the flagellar tip toward the cell body and vice-versa (red arrowheads); K = kinetoplasts; N = nucleus; black arrowheads = flagella; * = numerous pine-like pseudopods (acanthopodia) gave the cell a spiny appearance. Scale bar 10 μm.

permitted the intracellular conversion of the promastigotes to an amastigote-like state, and supported intracellular multiplication of *L. braziliensis*, *L. amazonensis*, *L. infantum*, and *L. major* (Fig 4). Cytosolic and intravascular intracellular localizations of the internalized parasite were observed. We found the parasites either membrane-bound within host cell–derived vacuoles (Fig 4A, 4B, 4C, 4D, 4E and 4H) or free in the cytosol (Fig 4F). The vacuoles showed either a tight-fitting (Fig 4E) or a loose membrane (Fig 4C and 4D) around the parasites.

### *Acanthamoeba–Leishmania* co-culture over different times of interaction

Moreover, we observed many vacuoles harboring several parasites, with the parasites attached or not attached to the wall (Fig 5, black arrows), and the amastigote partially adheres to the PV membrane with the remaining portion being free in vacuolar (Fig 5A). The amastigote forms are observed free in the cytosol (Fig 5B, 5L, and 5I; yellow arrows). Fig 5A and 5G show parasites within the vacuole had two nuclei and two kinetoplasts probably ready to undergo cellular division. This process suggests that amastigotes may replicate within amoeba, supported by the demonstration of duplicated kinetoplasts, nucleus (Fig 5C), and flagella (Fig 5G). The duplication of these organelles is known to represent the initial event of the parasite's mitosis. Moreover, we also observed morphologically intact parasites (Fig 5F and 5M), and or the rounded/oval body (Fig 5D and 5N) and started losing elongated shape towards a more rounded one (Fig 5D and 5E.). Fig 5P and 5C suggest that replicated amastigote forms were

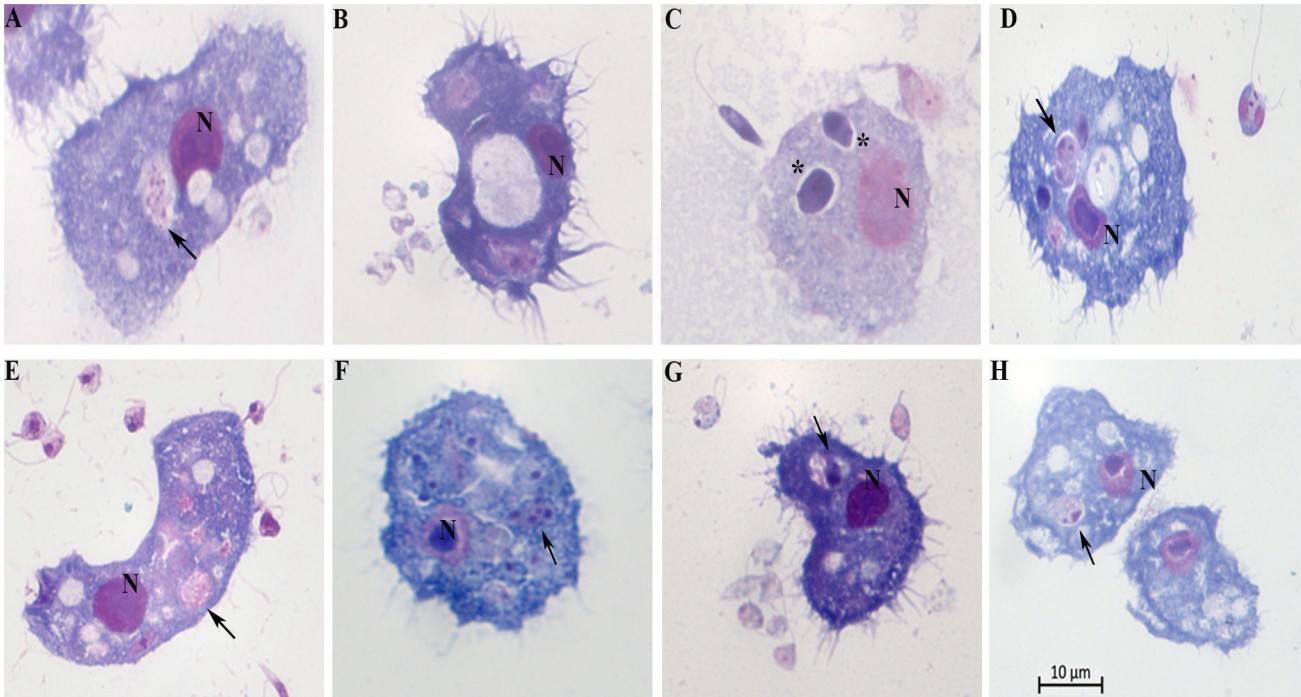

**Fig 4. Photomicrographs of *Acanthamoeba castellanii* infected with *Leishmania* spp., parasite ratio of: 10 for 3 h, stained with Panoptic fast staining.** A and B = *Leishmania. amazonensis;* C and D = *Leishmania. braziliensis*; E and F = *Leishmania infantum* and G and H = *Leishmania major.* N = nucleus of *A. castellanii*; black arrowheads indicate intracellular parasites inside vacuoles; * = the rounded/oval body; Scale bar 10 μm.

released, after the host cells burst or, alternatively, in a synchronized event reminiscent of exocytosis (red arrows). This speculation is purely based on the light microscopy and needs further observations.

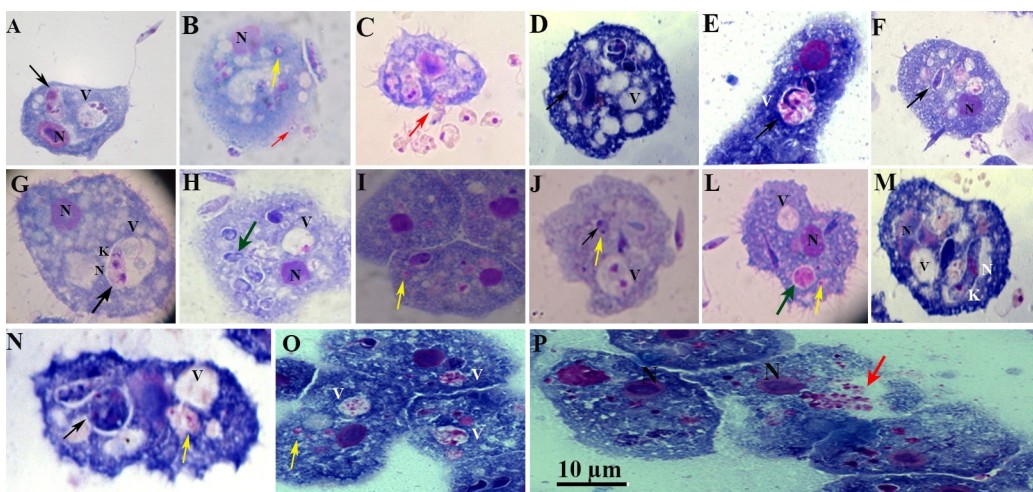

**Fig 5. Photomicrographs of *Acanthamoeba castellanii* infected with *Leishmania* spp., parasite ratio of 1:10 over different periods, stained with Panoptic fast staining** K = kinetoplast; N = nucleus; V = Vacuole; N = nucleus of the amoeba; black arrowheads indicate intracellular parasites inside vacuoles; Yellow arrowheads indicate amastigotes in the cytosol; red arrowheads indicate release of amastigotes; green arrowheads indicate amastigotes inside vacuoles; Scale bar 10 μm.

### Fluorescence microscopy images of *Acanthamoeba–Leishmania* co-culture at 24 h of interaction

Some of those observations were further confirmed by fluorescence microscopy imaging of *Acanthamoeba* infected with FITC-labeled promastigotes that reinforced our observation on internalization of *Leishmania* promastigotes. Fig 6 shows promastigotes undergoing significant changes, intact parasites (Fig 6C and 6D), rounded/oval bodies free in the cytosol (Fig 6A), or in vacuole (Fig 6C and 6D), that result in the obligatory amastigote-like intracellular forms, and fluorescence microscopy images of parasites multiplied within vacuoles (Fig 6C and 6D). Taken together, these results show that *Leishmania* promastigotes internalize and multiply into host cells.

Although different from each other, all the four infection profiles indicated that the interval for *Leishmania* is between 24 and 48h, as infected cells and the number of parasites per cell decreased after this period and in extra-cellular increased the number of rounded parasites with long flagella.

After *Acanthamoeba* lysis with 0.01% SDS [26], the release of amastigote forms was observed in the medium. These intracellular amastigotes were viable, since numerous newly generated free promastigote forms could be seen in the culture medium after transfer of 48

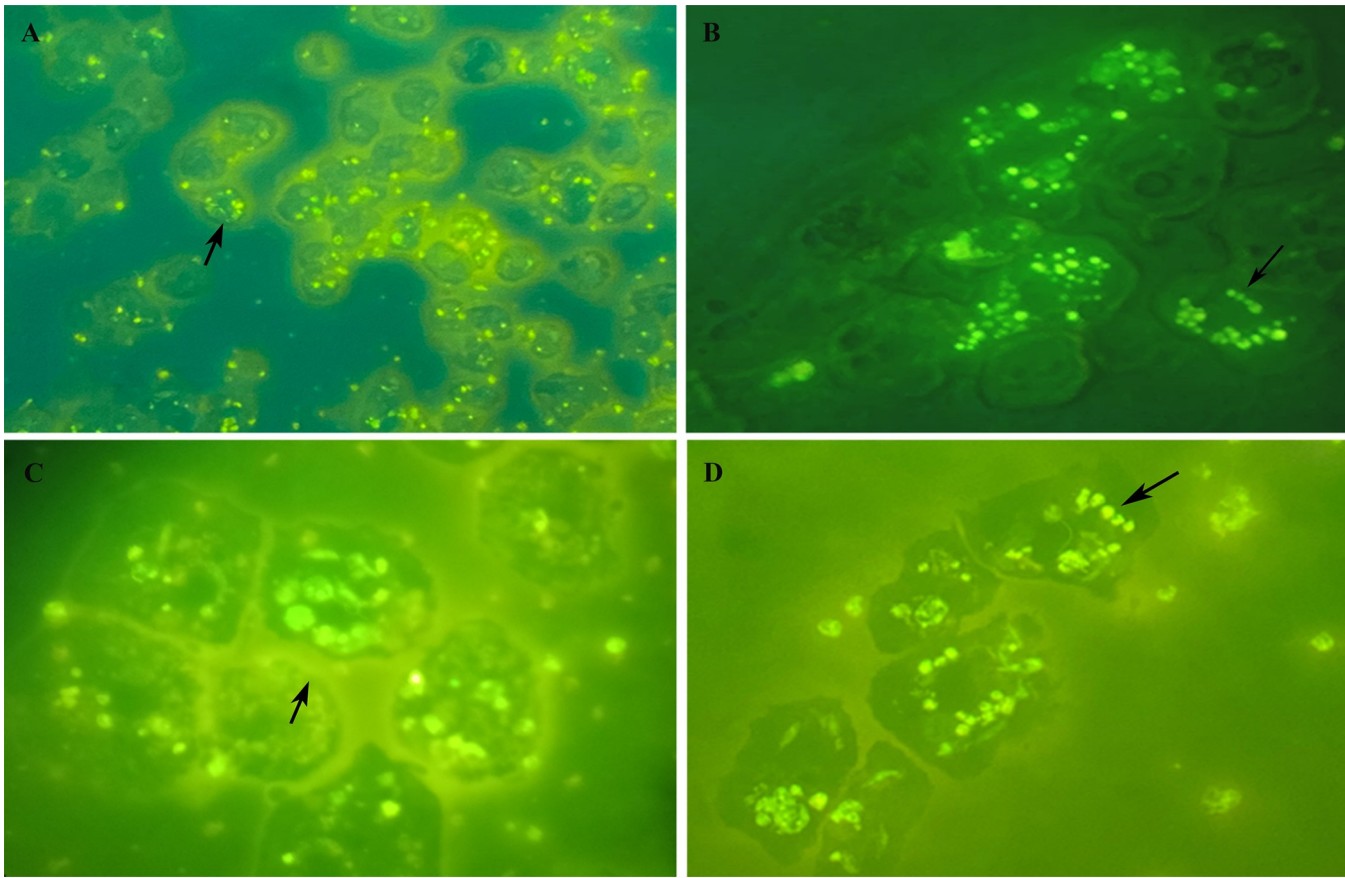

**Fig 6. Photomicrographs of *Acanthamoeba castellanii* infected with FITC-labeled promastigotes *Leishmania* spp., parasite ratio of 1:10 at 24 h of interactions.** (A) Low magnification view of *Leishmania*-FITC-labeled promastigotes, magnification × 100; (B) The fluorescence signal showing the labeling within trophozoite, magnification × 400; (C) Internalized promastigote and amastigotes, showing the labeling in vacuole, magnification × 1000, enlargement, and (D) illustrates the high degree of multiplication revealed by the labeling of intravacuolar amastigotes (black arrows), magnification × 1000.

and 72 h-cocultures for 4 to 6 additional days at 27°C. All results correspond to three independent assays run in duplicate using Schneider's medium at 27 or 37°C.

## Discussion

This report represents the first description of the abilities of *L. infantum*, *L. amazonensis*, *L. major*, and *L. braziliensis* to grow and subvert the functions of *A. castellanii* and *A. polyphaga*. Like macrophages, intracellular infection of *Acanthamoeba* spp. by *Leishmania* promastigotes relies on a series of key events, including attachment, internalization, amastigote differentiation, and intracellular survival [27]. The use of a cell line to study infections *in vitro* is an essential approach to evaluate the distinct aspects of *Leishmania* biology as well as its prevention and treatment, which require a clear understanding of the molecular network between parasites and their hosts [28]. Understanding the interactions would allow us to combat the disease by developing new drugs and therapies or by identifying important parasite antigens that could be exploited to develop vaccines.

There have been several reports of *Leishmania* parasitizing cells other than macrophages [29, 30]; however, it is recognized that in a vertebrate host, *Leishmania* spp. mainly resides within cells of the macrophage lineage [30]. Macrophages from different origins are amenable to successful *Leishmania* infection *in vitro* [31, 32]. These cells have been a mainstay of leishmaniasis research for more than 40 years, although there are many knowledge gaps. The main challenge is to elucidate that mechanisms lead to the subversion of macrophages [33]. Designing and applying a physiologically robust *in vitro* assay is highly relevant for drug screening and evaluation of biochemistry, physiology, and metabolism.

Macrophages represent the major effector cells that phagocytose parasites within vertebrate hosts [30]. The interaction between the innate and adaptive immune response is a crucial factor in the process of differentiating macrophages into pro-inflammatory (M1) or anti-inflammatory (M2) phenotypes, leading to resistance or susceptibility to *Leishmania* infection. Besides, various immune cells, including T cells, play a significant role in modulating macrophage polarization by releasing cytokines that influence macrophage maturation and function. Other immune cells can also influence macrophage polarization in a T cell–independent manner [32,33]. The host's immune responses and individual variations in parasite strains may be critical to determine the clinical outcome of leishmaniasis in a patient [31]. The influence of the innate immune response during *Leishmania* infection has been proposed to favor parasite survival, allowing it to fully establish itself within the mammalian host [30,34].

In the present study, *Acanthamoeba* spp. were similar to macrophages in that they were permissible to *Leishmania* spp., but they would have no influence on the adaptive immune response. Interestingly, *Acanthamoeba* spp. are well-characterized non-mammalian phagocytic cells with remarkable similarity to macrophages. Indeed, the similar traits and features of macrophages and *Acanthamoeba* spp.—including the ancient process of phagocytosis—provide evidence of a common ancestry [15,35] or convergent evolution. Mammalian macrophages and *Acanthamoeba* spp. display similarities in the molecular mechanisms involved in directional motility, recognition, binding, engulfment, and phagolysosome processes, and express common primitive receptors [16,17,36] Like macrophages, some microorganisms have evolved strategies for efficient uptake by *Acanthamoeba*, and they are able to regulate phagosome maturation to make the phagosome more hospitable for growth and to avoid destruction [19].

*Acanthamoeba* spp. are considered hosts for bacteria, viruses, yeast, and protozoa [37,38]. Their genome includes several genes that encode pattern recognition receptors (e.g., lipopolysaccharide-binding protein/bactericidal permeability enhancing protein [LBP/BPI], C-lectin,

C-type lectin, mannose-binding protein [MBP], D-galactoside/L-rhamnose-containing lectin SUEL domain, NB-ARC tetratricopeptide containing repeat-containing protein ERVR, and endogenous virus receptor) [39]. These receptors allow amoeba to interact with a vast array of species. Over the past several decades, *Acanthamoeba* spp. have received increasing attention due to their ability to act as a host/reservoir for microbes, including viruses (mimivirus, coxsackieviruses, adenoviruses, poliovirus, echovirus, enterovirus, vesicular stomatitis virus, etc.), bacteria (*Aeromonas*, *Bacillus*, *Bartonella*, *Burkholderia*, *Campylobacter*, *Chlamydophila*, *Coxiella*, *E. coli*, *Flavobacterium*, *Helicobacter*, *Legionella*, *Listeria*, *Staphylococcus*, *Mycobacterium*, *Pasteurella*, *Prevotella*, *Porphyromonas*, *Pseudomonas*, *Rickettsia*, *Salmonella*, *Shigella*, *Vibrio*, etc.), protists (*Cryptosporidium* and *Toxoplasma gondii*), and yeast (*Cryptococcus*, *Blastomyces*, *Sporothrix*, *Histoplasma*, *Streptomyces*, *Exophiala*, etc.) [16,17,40]. Moreover, several pathogens can survive within *A. castellanii* by using similar strategies as when they encounter macrophages [16,23,41–44]. A survey showed that the Dot/Icm type IV secretion system is required by *L. pneumophila* for intracellular proliferation within human macrophages and *Acanthamoeba* [43]. Moreover, *L. pneumophila* is taken up by *Acanthamoeba* by coiling phagocytosis but can impede phagosome–lysosome fusion. In turn, *Vibrio cholerae* can resist amoeboid phagosome killing by neutralizing proteolytic enzymes, changing the local pH, and counteracting reactive nitrogen and oxygen species that may otherwise kill internalized bacteria [45]. Likewise, invasion of *A. castellanii* or macrophages by *Mycobacterium* spp. shows notable similarities at the transcriptional and post-translational levels [46].

*A. castellanii* can also engulf fungi that are potential pathogens, such as *Cryptococcus neoformans*, *Histoplasma capsulatum*, *Blastomyces dermatitides*, *Fusarium solani*, and *Sporothrix schenckii* [47]. Mannose-binding lectins (MBLs) play a critical role in activating antifungal responses by immune cells [48,49]. A study has suggested that mannose-binding proteins identified in fungal recognition by both amoeboid and macrophage surfaces belong to the Conavalin A-like superfamily. Yet, the results reveal a limitation: a short sequence of 53 amino acids does not have enough resolution to predict protein folding accurately. [36]. However, data from the literature show us that macrophages and Acanthamoeba may share similar interaction mechanisms with fungal and bacteria pathogens, indicating a possible convergent evolution.

We found that *L. braziliensis*, *L. Amazonensis*, *L. infantum* and *L. major* could infect and grow within *A. castellanii* and *A. polyphaga*. These amoebae engulfed promastigotes, permitted the intracellular conversion of the promastigotes to an amastigote-like state, and supported their intracellular multiplication. There was a visible increase in the number of parasites per infected *Acanthamoeba* and in the total number of parasites per monolayer (although the number of parasites were not quantified). We evaluated attachment and infection primarily by inspecting fixed specimens from co-cultures of promastigotes and *Acanthamoeba* spp. A potential pitfall in this method is the possibility that some of the attached parasites might have been superimposed on the *Acanthamoeba* image and thus erroneously scored as phagocytosed. Despite this limitation, our findings provide novel insight into host–leishmania interactions.

*A. castellanii* showed slow growth over 6 days in LIT and Schneider's medium but remained viable and the death phase was not detected. Schneider's insect medium (Schneider's) and Grace's insect cell culture medium (Grace's) are defined media and are indicated for the continuous in vitro cultivation of promastigote forms of Leishmania [50]. However, a study demonstrated that the supplementation of LIT culture medium with fetal calf serum is the most suitable strategy to cultivate L. infantum parasites enabling the maintenance of growth and infective parasites for research uses. The authors showed that promastigote forms have a better growth in LIT and Schneider's with or without FCS when compared to that in Grace's [51].

In nature, promastigotes may be inoculated at a cutaneous site where complement is not active. *Acanthamoeba* spp. are lower eukaryotes that utilize a diverse repertoire of predicted

pattern recognition receptors, many with predicted orthologous functions in the innate immune systems of higher organisms [39]. Many studies have shown that the interaction between the innate and adaptive immune responses is a crucial factor leading to resistance or susceptibility to *Leishmania* infection [52]. It is conceivable that interfering with these processes represents a promising new strategy for antileishmanial intervention. In the present study, *Leishmania* spp. could subvert the functions of *A. castellanii* and *A. polyphaga* based on the innate immune response. *Leishmania* spp. could manipulate amoeboid functions to favor their survival and replication based on strategies of ancestral metabolic pathways. These novel findings indicate that *Leishmania* spp. and *Acanthamoeba* spp. co-cultures are a relatively inexpensive and easy-to-control model system for which multiple variables can be controlled. This cell model might provide new insights into how *Leishmania* infects cells and new therapeutic avenues to manipulate the innate immune response to protect cells from *Leishmania* infection.

## Conclusion

We have shown that *L. braziliensis*, *L. amazonensis*, *L. infantum*, and *L. major* are amoeba-resistant flagellates and that *A. castellanii* and *A. polyphaga* can be useful as a model to evaluate interactions for *Leishmania* species. Our findings contribute to the knowledge about the molecular aspects of the *Leishmania–Acanthamoeba* interaction considering innate immunity. The identification of promising molecules involved during phagocytosis and subversion of *Leishmania* spp. in amoeba could be extrapolated to macrophages. Thus, this cellular model should contribute to the understanding of the yet-to-be-elucidated mechanisms that are important to combat leishmaniasis.

## Supporting information

**S1 Raw Data. Raw data-Fig 2-The values used to build graphs about effect of media on the growth of *Acanthamoeba castellanii*.**
(XLSX)

## Author Contributions

**Conceptualization:** Helena Lúcia Carneiro Santos.

**Data curation:** Helena Lúcia Carneiro Santos.

**Funding acquisition:** Claudia Masini d'Avila.

**Methodology:** Helena Lúcia Carneiro Santos, Gabriela Linhares Pereira, Rhagner Bonono do Reis, Igor Cardoso Rodrigues.

**Resources:** Vitor Ennes Vidal.

**Writing – original draft:** Helena Lúcia Carneiro Santos.

**Writing – review & editing:** Helena Lúcia Carneiro Santos.

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
