## [Decision Letter · Decision Letter 0]

15 Jan 2024

Dear Dr. Santos,

Thank you very much for submitting your manuscript "Using Acanthamoeba spp. as a cell model to evaluate Leishmania infections." for consideration at PLOS Neglected Tropical Diseases. As with all papers reviewed by the journal, your manuscript was reviewed by members of the editorial board and by several independent reviewers. In light of the reviews (below this email), we would like to invite the resubmission of a significantly-revised version that takes into account the reviewers' comments. 

We cannot make any decision about publication until we have seen the revised manuscript and your response to the reviewers' comments. Your revised manuscript is also likely to be sent to reviewers for further evaluation.

Sincerely,

Naveed Ahmed Khan

Guest Editor

Ricardo Fujiwara

Section Editor

Reviewer's Responses to Questions

**Key Review Criteria Required for Acceptance?**

**Methods**

-Are the objectives of the study clearly articulated with a clear testable hypothesis stated?

-Is the study design appropriate to address the stated objectives?

-Is the population clearly described and appropriate for the hypothesis being tested?

-Is the sample size sufficient to ensure adequate power to address the hypothesis being tested?

-Were correct statistical analysis used to support conclusions?

-Are there concerns about ethical or regulatory requirements being met?

Reviewer #1: see below

Reviewer #2: the presented data are not clear and not conclusive, why using an amoebae instead of a macrophage?

**Results**

-Does the analysis presented match the analysis plan?

-Are the results clearly and completely presented?

-Are the figures (Tables, Images) of sufficient quality for clarity?

Reviewer #1: see below

Reviewer #2: the presented data are not clear and not conclusive, why using an amoebae instead of a macrophage?

Image quality is not good.

**Conclusions**

-Are the conclusions supported by the data presented?

-Are the limitations of analysis clearly described?

-Do the authors discuss how these data can be helpful to advance our understanding of the topic under study?

-Is public health relevance addressed?

Reviewer #1: The conclusions are well supported by the data

Reviewer #2: further work needs to be performed.

**Editorial and Data Presentation Modifications?**

Reviewer #1: (No Response)

Reviewer #2: NA

**Summary and General Comments**

Reviewer #1: Although Acanthamoeba have been isolated from the gut of wild Aedes aegypti mosquitoes (Otta at al, 2012) and the same species has been found to carry Leishmania spp. In Brazil (Coelho et al, 2017). It is not clear that Leishmania promastigotes are ever likely to encounter Acanthamoeba often. However, the present study aims to investigate the possible use of Acanthamoeba as a model system to study Leishmania promastigotes interaction with phagocytes. It is possible that by comparing the interaction of Leishmania promastigotes with Acanthamoeba and vertebrate derived macrophages important information may be obtained. Indeed, this has been fruitful in investigating the interactions of both Acanthamoeba and macrophages with Legionella bacteria (as the authors allude to). Perhaps the use of Dictyostelium discoideum would be even more productive given this organism’s famous genetic tractability and the large experimental tools available in this system?

Figure 2. It is unusual for each of the growth points to be labelled with the number of trophozoites? This makes the figure quite cluttered, and it would be much more helpful to include error bars here.

Line 348. This might be clearer as “Macrophages represent the major effector cells that phagocytose parasites within vertebrate hosts”

Line 397. It may not be accurate to describe the mannose binding proteins identified by de Souza Gonçalves et al, 2019 as being members of the ConA-like superfamily as there is only a 53 amino-acid sequence that is modelled to fold in a similar way to ConA? The word “ConA” should be spelled out in full as Concanavalin A anyway.

Line 251. “* = numerous pine-like pseudopods (acanthopodia)”. “spine” not “pine” in this sentence?

Line 412. Do the authors mean “remained viable and the death phase was not detected” rather than “analyzed” here?

Line 416. Italicise “L. infantum” 

Line 467. There are small problems with the reference list some of which are mentioned below but in addition there is in consistency in the use of italics for species names.

Line 497. Reference 9 is not formatted correctly, and some words run into each other.

Line 550. References 23 and 25 are the same paper.

Line 521. Missing “B” in author Bajgar name

Line 574. Reference 29 is not formatted correctly.

Line 670. Reference 52 is not formatted correctly.

Line 490 “quest for new drugs” not “que for new drugs” in the title of the Hefnawy paper.

Coelho, W. M. D., Bresciani, K. D. S., Coêlho, J. D. C. A., dos ANJOS, L. A., & Buzetti, W. A. S. (2017). Are Aedes aegypti mosquitoes potential vectors for leishmaniasis?–Case report. Brazilian Journal of Veterinary Research and Animal Science, 54(4), 416-419.

Otta DA, Rott MB, Carlesso AM, da Silva OS. Prevalence of Acanthamoeba spp. (Sarcomastigophora: Acanthamoebidae) in wild populations of Aedes aegypti (Diptera: Culicidae). Parasitol Res. 2012 Nov;111(5):2017-22. doi: 10.1007/s00436-012-3050-3. Epub 2012 Jul 25. PMID: 22828934.

Reviewer #2: the presented data are not clear and not conclusive, why using an amoebae instead of a macrophage?

PLOS authors have the option to publish the peer review history of their article (what does this mean?). If published, this will include your full peer review and any attached files.

Reviewer #1: Yes: Sutherland Maciver

Reviewer #2: No
---

## [Decision Letter · Decision Letter 1]

11 Jun 2024

Dear Dr. Santos,

Thank you very much for submitting your manuscript "Using Acanthamoeba spp. as a cell model to evaluate Leishmania infections." for consideration at PLOS Neglected Tropical Diseases. As with all papers reviewed by the journal, your manuscript was reviewed by members of the editorial board and by several independent reviewers. In light of the reviews (below this email), we would like to invite the resubmission of a significantly-revised version that takes into account the reviewers' comments. 

Although the revised manuscript has improved but as you will see that reviewers have raised concerns. I do feel that there is merit in your study but I would urge you to carefully address these comments in as much detail as possible to ensure reviewer is satisfied with the revision.

We cannot make any decision about publication until we have seen the revised manuscript and your response to the reviewers' comments. Your revised manuscript is also likely to be sent to reviewers for further evaluation.

Sincerely,

Naveed Ahmed Khan

Guest Editor

Ricardo Fujiwara

Section Editor

Although the revised manuscript has improved but as you will see that reviewers have raised concerns. I do feel that there is merit in your study but I would urge you to carefully address these comments in as much detail as possible to ensure reviewer is satisfied with the revision.

Reviewer's Responses to Questions

**Key Review Criteria Required for Acceptance?**

**Methods**

-Are the objectives of the study clearly articulated with a clear testable hypothesis stated?

-Is the study design appropriate to address the stated objectives?

-Is the population clearly described and appropriate for the hypothesis being tested?

-Is the sample size sufficient to ensure adequate power to address the hypothesis being tested?

-Were correct statistical analysis used to support conclusions?

-Are there concerns about ethical or regulatory requirements being met?

Reviewer #1: yes

Reviewer #3: The study aims to determine where Acanthamoeba is a suitable, alternative model to macrophages. The reasons for this new model are not clear as the in vitro macrophages model is suitable already. However, Acanthamoeba may be more cost effective, but this is not mentioned in the manuscript. There is suggestion that Acanthamoeba 

The study design could be developed further. It is limited to observation only and there is not mechanism described. It would be useful to have include a macrophage control to compare and contrast behavior. I would also suggest further characterization using confocal or electron microscopy. There is suggestion that innate immune systems are similar between Acanthamoeba and macrophages, but this has not been presented in detail. Further experimentation is needed to develop this hypothesis. I think that reference to innate immune system might only refer to the process of phagocytosis. This should be clarified. No ethical concerns, but health and safety should be observed. 

Is there a reference of the use of 0.01% SDS to break open amoebae?

**Results**

-Does the analysis presented match the analysis plan?

-Are the results clearly and completely presented?

-Are the figures (Tables, Images) of sufficient quality for clarity?

Reviewer #1: yes

Reviewer #3: The results are limited to microscopic observations. Figure 3 is useful but hard to follow as it is not clear if Leishmania are actually inside the amoeba. 

It would be useful to have subheadings in the results and it was difficult to follow with the figure legends in the middle of the text.

**Conclusions**

-Are the conclusions supported by the data presented?

-Are the limitations of analysis clearly described?

-Do the authors discuss how these data can be helpful to advance our understanding of the topic under study?

-Is public health relevance addressed?

Reviewer #1: yes

Reviewer #3: The conclusions are only partly supported by the data. Acanthamoeba can host Leishmania but there is not sufficient understanding of the molecular aspects of the Leishmania–Acanthamoeba interaction considering innate immunity. the public health relevance is not elaborated and due to this I am not sure if this is a relevant journal for this study?

**Editorial and Data Presentation Modifications?**

Reviewer #1: (No Response)

Reviewer #3: Line 150: change parasite to amoeba

Line 159: specie should be species 

Line 174: why these temperatures? 

Line 194: has 0.01%SDS been used before? please reference

in result section I would suggest the use of subheadings

Lines 224-227: review grammar

Line 225: Acanthamoeba move very very slowly. It would not be possible to see them move without a time lapse. Can this be explained? 

Line 269: what is meant by 'loose vacuole'?

Lines 425-428 I don't understand the reference to the innate immune system from this sentence. There is not study of innate immune system

**Summary and General Comments**

Reviewer #1: The authors have addressed the points made in my review of the earlier version of this manuscript. I know think this paper is ready for publication.

Reviewer #3: The study is novel, but I am not sure of the relevance to the 'real world' and as a model for Leishmania-macrophage interactions, seeing as we already have a good model in vitro. 

I can see that this study provides a good foundation to understanding microbial interactions. Seeing as there are other amoebae that harbor kinetoplastid endosymbionts, this is a good foundation to understanding how this relationship has developed. Therefore, I would ask if PNTD is the most appropriate journal for this study.

PLOS authors have the option to publish the peer review history of their article (what does this mean?). If published, this will include your full peer review and any attached files.

Reviewer #1: Yes: Sutherland Maciver

Reviewer #3: No
---

## [Decision Letter · Decision Letter 2]

6 Sep 2024

Dear Dr. Santos,

We are pleased to inform you that your manuscript 'Using Acanthamoeba spp. as a cell model to evaluate Leishmania infections.' has been provisionally accepted for publication in PLOS Neglected Tropical Diseases.

Best regards,

Naveed Ahmed Khan

Guest Editor

Abhay Satoskar

Section Editor

Your revision and responses are satisfactory to reviewers. I hope that you will agree that the extensive changes have improved the manuscript. The manuscript is acceptable.

Reviewer's Responses to Questions

**Key Review Criteria Required for Acceptance?**

**Methods**

-Are the objectives of the study clearly articulated with a clear testable hypothesis stated?

-Is the study design appropriate to address the stated objectives?

-Is the population clearly described and appropriate for the hypothesis being tested?

-Is the sample size sufficient to ensure adequate power to address the hypothesis being tested?

-Were correct statistical analysis used to support conclusions?

-Are there concerns about ethical or regulatory requirements being met?

Reviewer #4: (No Response)

**Results**

-Does the analysis presented match the analysis plan?

-Are the results clearly and completely presented?

-Are the figures (Tables, Images) of sufficient quality for clarity?

Reviewer #4: (No Response)

**Conclusions**

-Are the conclusions supported by the data presented?

-Are the limitations of analysis clearly described?

-Do the authors discuss how these data can be helpful to advance our understanding of the topic under study?

-Is public health relevance addressed?

Reviewer #4: (No Response)

**Editorial and Data Presentation Modifications?**

Reviewer #4: (No Response)

**Summary and General Comments**

Reviewer #4: (No Response)

PLOS authors have the option to publish the peer review history of their article (what does this mean?). If published, this will include your full peer review and any attached files.

Reviewer #4: **Yes: **David Lloyd

---

## [Editor Report · Acceptance letter]

26 Sep 2024

Dear Dr. Santos,

We are delighted to inform you that your manuscript, "Using Acanthamoeba spp. as a cell model to evaluate Leishmania infections.," has been formally accepted for publication in PLOS Neglected Tropical Diseases.

Best regards,

Shaden Kamhawi

co-Editor-in-Chief

Paul Brindley

co-Editor-in-Chief
